Two new syntopic species of glassfrogs (Amphibia, Centrolenidae, Centrolene) from the southwestern Andes of Ecuador

http://orcid.org/0000-0002-6132-2738 Cisneros-Heredia Diego F. 1 2 diego.cisnerosheredia@gmail.com
http://orcid.org/0000-0003-3224-1987 Yánez-Muñoz Mario H. 1
http://orcid.org/0000-0001-7971-1216 Sánchez-Nivicela Juan C. 1 2 3
http://orcid.org/0000-0001-6300-9350 Ron Santiago R. 4
1 División de Herpetología, Instituto Nacional de Biodiversidad , Quito , Ecuador
2 Laboratorio de Zoología Terrestre, Instituto de Biodiversidad Tropical IBIOTROP, Colegio de Ciencias Biológicas y Ambientales, Universidad San Francisco de Quito USFQ , Quito , Ecuador
3 Facultad de Ciencias, Grupo de Investigación en Evolución y Ecología de Fauna Neotropical, Universidad Nacional de Colombia , Bogotá D.C. , Colombia
4 Escuela de Biología, Museo de Zoología, Pontificia Universidad Católica del Ecuador , Quito , Ecuador
Morrone Juan J.
Electronic publication date: 2023 May 9
Publication date: 2023
Volume: 11
Electronic Location ID: e15195
Received 2023 Jan 19; Accepted 2023 Mar 15
Copyright: © 2023 Cisneros-Heredia et al.
Copyright year: 2023
Copyright holder: Cisneros-Heredia et al.
License: This is an open access article distributed under the terms of the Creative Commons Attribution License, which permits unrestricted use, distribution, reproduction and adaptation in any medium and for any purpose provided that it is properly attributed. For attribution, the original author(s), title, publication source (PeerJ) and either DOI or URL of the article must be cited.
License URL: https://creativecommons.org/licenses/by/4.0/

Keywords: Andes, Anura, Centrolene camposi sp. nov., Centrolene condor, Centrolene ericsmithi sp. nov., New species, Phylogenetic relationships, Taxonomy

Funding: INABIO Secretaría Nacional de Educación Superior, Ciencia, Tecnología e Innovación del Ecuador SENESCYT Pontificia Universidad Católica del Ecuador, Dirección General Académica Universidad San Francisco de Quito USFQ (COCIBA grants, IBIOTROP operating funds) Gobierno Autónomo Provincial de El Oro financed field and lab work through the project “Amphibians, Reptiles and Birds of the El Oro Province” implemented by the INABIO. Laboratory work in Ecuador was funded by Secretaría Nacional de Educación Superior, Ciencia, Tecnología e Innovación del Ecuador SENESCYT (Arca de Noé initiative; Santiago R Ron and Omar Torres principal investigators) and by a grant from Pontificia Universidad Católica del Ecuador, Dirección General Académica. Work by Diego F Cisneros-Heredia was funded by Universidad San Francisco de Quito USFQ (COCIBA grants, IBIOTROP operating funds). The funders had no role in study design, data collection and analysis, decision to publish, or preparation of the manuscript.

==============================
We describe two new species of glassfrogs of the genus Centrolene living in syntopy at La Enramada, province of Azuay, southwestern Ecuador. They were found in a small creek in montane evergreen forests at 2,900 m elevation. The first new species is distinguished from all other members of the genus Centrolene by having the following combination of characters: dentigerous process of vomer absent; sloping snout in lateral view; thick, white labial stripe and a faint white line between the lip and anterior ¼ of body; humeral spine in adult males; parietal peritoneum covered by iridophores, visceral peritonea translucent (except pericardium); ulnar and tarsal ornamentation; dorsal skin shagreen with dispersed warts; uniform green dorsum with light yellowish green warts; and green bones. The new species is remarkable by being sister to a species from the opposite Andean versant, C. condor. The second new species is distinguished from all other Centrolene by having the following combination of characters: dentigerous process of vomer absent; round snout in lateral view; thin, yellowish labial stripe with a row of white tubercles between the lip and arm insertion, and a yellowish line between arm insertion and groin; uniform green dorsum; humeral spine in adult males; parietal peritoneum covered by iridophores, visceral peritonea translucent (except pericardium); dorsal skin shagreen with dispersed spicules; ulnar and tarsal ornamentation; and green bones. The second new species is sister to C. sabini and an undescribed species of Centrolene from southeastern Ecuador. Based on nuclear and mitochondrial DNA sequences, we present a new phylogeny for Centrolene and comment on the phylogenetic relationships inside the genus.

Introduction

Glassfrogs of the genus Centrolene Jiménez de la Espada, 1872 are distributed across the Andes, from the Merida Massif in Venezuela to the Kosñipata Valley in southern Peru (Frost, 2021). While no morphological synapomorphies are known for Centrolene, its monophyly is well-supported (Guayasamin et al., 2009, 2020; Catenazzi et al., 2012; Twomey, Delia & Castroviejo-Fisher, 2014). The following combination of morphological characters is helpful to diagnose Centrolene: presence of humeral spines in adult males of most species—except Centrolene daidalea (Ruiz-Carranza & Lynch, 1991a) and C. savagei (Ruiz-Carranza & Lynch, 1991a); liver lobed and covered by translucent hepatic peritoneum; pericloacal warts enamelled; bones green in life; background colouration of dorsum in preservative lavender (Ruiz-Carranza & Lynch, 1991b; Cisneros-Heredia & McDiarmid, 2007; Guayasamin et al., 2009; Catenazzi et al., 2012).

Twelve species of Centrolene have been reported from Ecuador: C. ballux (Duellman & Burrowes, 1989); C. buckleyi (Boulenger, 1882); C. charapita Twomey, Delia & Castroviejo-Fisher, 2014; C. condor (Cisneros-Heredia & Morales-Mite, 2008); C. geckoidea Jimenez de la Espada, 1872; C. heloderma (Duellman, 1981); C. huilensis (Ruiz-Carranza & Lynch, 1995), C. lynchi (Duellman, 1980), C. medemi (Cochran & Goin, 1970), C. peristicta (Lynch & Duellman, 1973), C. pipilata (Lynch & Duellman, 1973), and C. sanchezi (Ruiz-Carranza & Lynch, 1991c; Lynch & Duellman, 1973; Cisneros-Heredia & McDiarmid, 2005, 2006; Cisneros-Heredia & Yánez-Muñoz, 2007a; Cisneros-Heredia & Morales-Mite, 2008; Guayasamin et al., 2020). Six of them inhabit the north-western slopes of the Cordillera Occidental of the Andes of Ecuador: C. ballux, C. buckleyi, C. geckoidea, C. heloderma, C. lynchi and C. peristicta. Still, only C. heloderma has been reported from the southwestern slopes. Yánez-Muñoz (2015) preliminarily informed of the presence of C. heloderma in the southwestern Andes of Ecuador based on three specimens collected at La Enramada, province of Azuay. However, molecular analyses show that they belong to two different and undescribed species of Centrolene found together at one of the region’s last remnants of montane forests. We are pleased to describe these two new species in this publication.

Materials and Methods

Ethics statement

Our study was authorised under framework contracts for access to genetic resources MAE-DNB-CM-2016-0045 and MAE-DNB-CM-2019-0120, issued by the Ministerio del Ambiente del Ecuador. We followed the standard guidelines for using live amphibians and reptiles in field research by Beaupre et al. (2004).

Species concept

We consider species as separately evolving metapopulation lineages, recognisable from an operational point of view to the extent that isolation from their putative sister lineages can be inferred (De Queiroz, 2007).

Taxonomic sampling

Specimens from the following collections were examined: División de Herpetología, Museo Ecuatoriano de Ciencias Naturales, Instituto Nacional de Biodiversidad, Quito (DHMECN); University of Kansas Natural History Museum, Lawrence (KU); Museo de Zoología, Pontificia Universidad Católica del Ecuador, Quito (QCAZ); National Museum of Natural History, Smithsonian Institution, Washington, D.C. (USNM).

Information on species for comparative diagnoses was obtained from the literature (Duellman & Schulte, 1993; Señaris & Ayarzaguena, 2005; Cisneros-Heredia & McDiarmid, 2007; Catenazzi et al., 2012; Twomey, Delia & Castroviejo-Fisher, 2014; Guayasamin et al., 2020) and the following examined specimens: Centrolene ballux (12 specimens): ECUADOR: province of Carchi: 5 km W of La Gruel (KU 202798); province of Pichincha: Las Gralarias (QCAZ 40195–97); 14 km W of Chiriboga (KU 164726–32); Quebrada Zapadores (KU 164733). C. buckleyi (44 specimens): ECUADOR: province of Bolívar: Guanujo (DHMECN 0866–67); province of Carchi: Los Encinos (DHMECN 1246); Cabaña Las Orquídeas Morán (DHMECN 13375, 13376, 13828, 14180); province of Cotopaxi: Pilalo (USNM 288428); province of Napo: Santa Bárbara (USNM 311113–14); province of Pichincha: Quito (USNM 288423); 8.5 km (by road) NW of Nono (USNM 286626–27); Machachi (USNM 286628–29); 21.2 km (by road) ESE of Chiriboga (USNM 286630–31); 8 km to Chiriboga (USNM 288424); province of Sucumbíos: near Santa Bárbara (DHMECN 868–893). C. condor (seven specimens): ECUADOR: province of Zamora Chinchipe: Destacamento Militar Cóndor Mirador (QCAZ 37279); Paquisha Alto (DHMECN 11208–11210); Concesión Colibrí (DHMECN 12049); Concesión La Zarza (DHMECN12053); province of Morona-Santiago: near Reserva Biológica El Quimi (QCAZ 72514). C. heloderma (11 specimens): ECUADOR: province of Pichincha: Quebrada Zapadores (USNM 211219–21); 13.1 km NW of Nono (USNM 211216–7); 8.6 km SE of Tandayapa (USNM 211218); Reserva Las Gralarias (QCAZ 40200, 50722); 14 km W of Chiriboga (QCAZ 44881); province of Carchi: Reserva Dracula, El Guapilal (DHMECN 14999–15000). Additional specimens examined during our studies in Centrolenidae are listed in Cisneros-Heredia & McDiarmid (2007) and Guayasamin et al. (2020).

Fieldwork

Fieldwork was conducted at La Enramada (3.161074 °S; 79.600045 °W, 2,900 m), province of Azuay, Ecuador, during expeditions of the Instituto Nacional de Biodiversidad INABIO on 21–31 March 2015, 13–17 April 2019 and 06–11 December 2022. We used visual encounter surveys for herpetological searches (Crump & Scott, 1994). Only the first expedition in March 2015 resulted in the collection of specimens of the new species described herein. Individuals were photographed alive and euthanised with benzocaine, a muscle tissue sample was extracted and preserved in 95% ethanol, and whole specimens were fixed in 10% formalin and preserved in 75% ethanol.

Morphology and colouration

Diagnosis, terminology, and adult characters and measurements follow the format and definitions proposed by Cisneros-Heredia & McDiarmid (2007). All characteristics reported in the description of the type series are from adult specimens. Sex and maturity were determined by directly examining gonads through dissections and noting the presence of secondary sexual characters (i.e., vocal slits and nuptial pads). All morphometric data were measured with a digital calliper (0.05 mm accuracy, rounded to the nearest 0.1 mm) under a stereomicroscope, reported as a range (mean ± standard deviation), and included: snout-vent length (SVL), head length (HL), head width (HW), interorbital distance (IOD), eye diameter (ED), internarial distance (IND), eye-nostril distance (EN), tympanum diameter (TD), tibia length (TL), foot length (FL), hand length (HAL), Finger III disk width (Fin3DW). Colour patterns are described based on photographs of live specimens taken in the field. The adjective “enamelled” describes the shiny white colouration produced by an accumulation of iridophores (Lynch & Duellman, 1973; Cisneros-Heredia & McDiarmid, 2007).

Phylogenetic analyses and genetic distances

To assess the evolutionary relationships of the new species, we sequenced three mitochondrial genes (12S rRNA, 16S rRNA, and NADH dehydrogenase subunit) and two nuclear genes (RAG1 and C-MYC 2). DNA was extracted from muscle or liver tissue preserved in 95% ethanol using standard phenol-chloroform extraction protocols (Sambrook, Fritsch & Maniatis, 1989). PCR amplification was performed under standard protocols and sequenced by the Macrogen Sequencing Team (Macrogen Inc., Seoul, Korea). We also added a short new sequence of C. lynchi QCAZ 40192 (3′ end of 16S, tRNA-Leu, and 5′ beginning of ND1) because in a preliminary phylogeny, C. lynchi GenBank sequences QCAZ 40192 and QCAZ 40191, from the same population, unexpectedly, were non-monophyletic. Upon further inspection, we realised they lacked overlapping sequences. The new sequence of QCAZ 40192 solved that problem.

Our phylogeny is based on sequences of Centrolene from GenBank (published by Guayasamin et al., 2008, 2020; Castroviejo-Fisher et al., 2014; Twomey, Delia & Castroviejo-Fisher, 2014) and the newly generated sequences (see above). We analysed the mitochondrial genes 12S rRNA, 16S rRNA, ND1 and the nuclear genes BDNF, C-MYC 2, CXCR4, POMC, RAG1, SLC8A1, SLC8A3, for a total of 10 loci and up to 6,355 bp. We also included Genbank sequences of species of Allophryne, Celsiella, Chimerella, Cochranella, Espadarana, Hyalinobatrachium, Ikakogi, Nymphargus, Rulyrana, Sachatamia, Teratohyla, and Vitreorana. The phylogeny was rooted with Allophryne ruthveni (specimen MAD1857; outgroup choice based on Twomey, Delia & Castroviejo-Fisher, 2014). The matrix had 61 terminals. GenBank accession numbers for newly generated sequences are in Table 1.

Table 1 Species, vouchers, and GenBank accession numbers for newly generated DNA sequences used in genetic analyses.

Species, vouchers, and GenBank accession numbers for newly generated DNA sequences used in genetic analyses. Acronyms are DHMECN = División de Herpetología, Museo Ecuatoriano de Ciencias Naturales, Instituto Nacional de Biodiversidad, Quito; QCAZ = Museo de Zoología, Pontificia Universidad Católica del Ecuador.

Museum voucher	Species	Locality	12S	16S	ND1	RAG1	C-MYC 2	
DHMECN10221/QCAZ59064	C. sanchezi	Ecuador: Zamora Chinchipe: Cordillera del Condor	OQ225626	–	OQ248672	OQ248679	OQ248668	
DHMECN10222/QCAZ59065	C. sanchezi	Ecuador: Zamora Chinchipe: Cordillera del Condor	OQ225627	–	OQ248673	OQ248680	OQ248669	
DHMECN11406/QCAZ59073	C. ericsmithi	Ecuador: Azuay: San Rafael, la Enramada	OQ225628	–	OQ248674	OQ248682	OQ248670	
DHMECN11407/QCAZ59070	C. camposi	Ecuador: Azuay: San Rafael, la Enramada	OQ225629	OQ225616	OQ248675	OQ248681	–	
DHMECN11408/QCAZ59076	C. camposi	Ecuador: Azuay: San Rafael, la Enramada	–	–	OQ248676	OQ248683	–	
QCAZ65013	C. sanchezi	Ecuador: Zamora Chinchipe: Reserva Biologica Cerro Plateado	OQ225630	–	OQ248677	–	OQ248671	
QCAZ72514	C. condor	Ecuador: Morona Santiago: Reserva Biologica El Quimi	–	OQ225617	OQ248678	–	–	
QCAZ40192	C. lynchi	Ecuador: Pichincha: Reserva las Gralarias	–	To be added	–	–	–	

Raw sequences were assembled with Geneious 9.1.8 software (Biomatters Ltd., Auckland, NI, NZ). Sequences were aligned using MAFFT 7.017 and the L-INS-I algorithm (Katoh & Standley, 2013). The alignment was visually inspected in Mesquite (version 3.61; Maddison & Maddison, 2019), and alignment errors were manually adjusted. We partitioned the matrix to allow separate evolution models for each gene and codon position (except for 12S and 16S non-coding) for a total of 26 partitions. We accomplished that with the command-m MPF (Chernomor, von Haeseler & Minh, 2016; Kalyaanamoorthy et al., 2017) in the software IQ-TREE multicore version 2.2.0 (Minh et al. 2020). The phylogeny was estimated under maximum likelihood using IQ-TREE 2.2.0 under default settings. To assess branch support, we made 200 non-parametric bootstrap searches (-b 200 command) and 1,000 replicates for the SH-like approximate likelihood ratio test (-alrt 1,000 command); (Guindon et al. 2010). We considered that branches with bootstrap values >70 and SH-aLRT values >80 had strong support. Pairwise uncorrected p-genetic distances were calculated using the software MEGA 11.0.13 (Tamura, Stecher & Kumar, 2021). The standard error of the genetic distance was estimated with the bootstrap method. For accuracy, we only compared overlapping fragments longer than 400 bp.

Nomenclatural acts

The electronic version of this article in Portable Document Format (PDF) will represent a published work according to the International Commission on Zoological Nomenclature (ICZN). Hence the new names contained in the electronic version are effectively published under that Code from the electronic edition alone. This published work and its nomenclatural acts have been registered in ZooBank, the online registration system for the ICZN. The ZooBank LSIDs (Life Science Identifiers) can be resolved, and the associated information viewed through any standard web browser by appending the LSID to the prefix http://zoobank.org/. The LSID for this publication is urn:lsid:zoobank.org:pub:A2A88B00-DA2C-443E-BC8B-9922980F8789. The online version of this work is archived and available from the following digital repositories: PeerJ, PubMed Central and CLOCKSS.

Results

Phylogenetic relationships

Our phylogenetic tree (Fig. 1) is generally consistent with previous phylogenetic analyses of Centrolenidae (e.g., Twomey, Delia & Castroviejo-Fisher, 2014; Guayasamin et al., 2020). Unlike Guayasamin et al. (2020), we found a clade that excludes C. charapita and C. geckoidea and unites two sister subclades: (C. savagei + (C. daidalea + C. sp. Ca01)) + (C. antioquiense + C. peristicta) and a clade containing all remaining species of Centrolene. The two new species described herein belong to the later subclade. We included two specimens of C. condor in our phylogeny. We identify specimen QCAZ 72514 as C. condor based on its morphology, colouration, and distribution—it was collected near the species’ type locality. Specimen QCAZ 44896 is a tadpole and was reported as an undescribed species by Guayasamin et al. (2020), but it is closely related to QCAZ 72514 and is herein reported as C. condor. Specimen QCAZ 47338, reported as C. condor by Guayasamin et al. (2020), is considered an undescribed species.

Figure 1 Maximum likelihood tree of Centrolene inferred from a partitioned analysis of 6,355 aligned sites of DNA sequences of 10 nuclear and mitochondrial loci.

SH-aLRT support (before the slash) and non-parametric bootstrap support (after) are shown as percentages on branches. The specimen voucher number precedes the species name; if a voucher number was unavailable, we added the GenBank accession number after the species name. Outgroups are not shown.

The first new species is strongly supported as sister to C. condor, a species only known from the Cordillera del Cóndor, southeastern Ecuador. The uncorrected-p genetic distance between them is 1.04% (SE = 0.338%) for the gene 12S. In Centrolene, at least two pairs of sister species are separated by distances (gene 12S) lower than 1%: C. altitudinale vs C. notosticta (0.8%) and C. peristicta vs. C. antioquiense (0.6–0.7%). Therefore, the 12S genetic distance between the first species and C. condor falls within the observed range of interspecific distances for the genus. The genetic distance between C. condor and the first new species for ND1 ranges from 6.1% (SE = 0.786%) to 6.5% (SE = 0.818%). The second new species is sister to a clade composed of C. sabini (from southeastern Peru) and an undescribed species of Centrolene from southeastern Ecuador (MRy 547), referred to as [Ca1] by Amador et al. (2018). The uncorrected p-genetic distance (12S) between the second new species and C. sabini is 2.9% (SE = 0.549%), while the distance with Centrolene sp. (MRy 547) is 3.7% (SE = 0.709%).

Species descriptions

Centrolene camposi sp. nov.

LSID urn:lsid:zoobank.org:act:868316B5-0ED5-4A21-AE3A-0488D98E418B

(Figs. 2–6)

Figure 2 Holotypes of the new species.

(A) Centrolene camposi sp. nov. in life, male, DHMECN 11407; and (B) holotype of Centrolene ericsmithi sp. nov. male, DHMECN 11406. Photographs by Juan Carlos Sánchez-Nivicela.

Figure 3 Dorsal, ventral, and dorsolateral views of the type specimens of the new species shortly after euthanasia.

(A–F) Holotype of Centrolene camposi sp. nov., adult male, DHMECN 11407. (G–K) Holotype of Centrolene ericsmithi sp. nov., adult male, DHMECN 11406. HS, humeral spines. Photographs by Juan Carlos Sánchez-Nivicela.

Figure 4 Comparison of the new species and their closely related lineages, in dorsal view.

(A) Centrolene camposi sp. nov. male holotype, DHMECN 11407; (B) C. ericsmithi sp. nov., male, DHMECN 11406; (C) C. heloderma, male, DHMECN 15000; (D) C. condor, male, DHMECN 11240; (E) C. lynchi, male, QCAZ 40194; (F) C. buckleyi, male, DHMECN 13375. Photographs by Mario H. Yánez-Muñoz.

Figure 5 Comparison of the new species and their closely related lineages.

First column: head in dorsal view; second column: head in ventral view; third column: head in lateral view. (A–C) Centrolene camposi sp. nov. male holotype, DHMECN 11407; (D–F) C. condor, male, DHMECN 11240; (G–I) C. ericsmithi sp. nov., male, DHMECN 11406; (J–L) C. lynchi, male, QCAZ 40194; (M–O) C. heloderma, male, DHMECN 15000; (P–R) C. buckleyi, male, DHMECN 13375. Photographs by Mario H. Yánez-Muñoz.

Figure 6 Comparison of the new species and their closely related lineages, in life in dorsolateral view.

(A) Centrolene camposi sp. nov. male holotype, DHMECN 11407; (B) C. ericsmithi sp. nov., male holotype, DHMECN 11406; (C) C. heloderma, male, QCAZ 50722; (D) C. condor, male, QCAZ 72520; (E) C. lynchi, male, not collected; (F) C. buckleyi, male, QCAZ 40308. Photographs by Juan Carlos Sánchez-Nivicela (A, B), Santiago R. Ron (C–E), and Mario H. Yánez-Muñoz (E).

Centrolene heloderma Yánez-Muñoz (2015)

Centrolene sp. 2. Bejarano-Muñoz, Sánchez-Nivicela & Yánez-Muñoz (2019)

Proposed Spanish common name. Rana de Cristal de Campos

Proposed English common name: Campos’ Glassfrog

Holotype. DHMECN 11407 (field number 3566), adult male (Figs. 2 and 3) from La Enramada (3.1628°S; 79.5886°W, 2,950 m), provincia de Azuay, República del Ecuador, collected by J. Sánchez-Nivicela on 31 March 2015.

Paratype. DHMECN 11408, adult male, same data as holotype.

Definition. Centrolene camposi sp. nov. is distinguished from all other Centrolenidae by the following combination of characters: (1) dentigerous process of vomer absent; (2) snout rounded to subacuminate in dorsal view, sloping in lateral view; (3) tympanic annulus barely visible, lower ¾ visible, tympanic membrane coloured as dorsal skin, supratympanic fold present and low; (4) dorsal skin shagreen with dispersed low and rounded warts, and microspicules and spicules present (at least in males); (5) ventral skin granular, subcloacal area enamelled, strongly granular with two large subcloacal warts and with enamelled cloacal sheath; (6) parietal peritoneum white, iridophores covering 2/3 the parietal peritoneum (condition P3); pericardium covered by iridophores, all other visceral peritonea clear (condition V1); (7) liver lobed (five lobes) and hepatic peritoneum clear (lacking iridophore layer, condition H0); (8) adult males with projecting humeral spine; (9) basal webbing between fingers I and II, moderate webbing between fingers II and IV, II (2−–2)–3+ III 2½–2+ IV; (10) toe webbing I (1––1½)–(2–2+) II (0+–1−)–(2½–2+) III (1+–1½)–2½ IV 2½–1½ V; (11) low, enamelled metacarpal fold continuing with elevated, thick, enamelled ulnar fold; elevated, low, enamelled metatarsal and tarsal fold; low tarsal fringe on inner tarsus; (12) nuptial excrescences type I; concealed prepollex; (13) Finger I shorter than Finger II; (14) diameter of eye larger than width of disc on Finger III; (15) colour in life, bright green dorsum, thick yellowish-white labial stripe continuing into a faint yellowish line between lip and anterior ¼ of body, yellowish green flanks, hidden surfaces of limbs and digits, enamelled metacarpal, ulnar, metatarsal and tarsal folds, bones green; (16) colour in preservative, lavender dorsum with translucent spicules, enamelled labial stripe continuing into a faint enamelled line between lip and anterior ¼ of body, faint enamelled metacarpal, ulnar, metatarsal and tarsal folds; (17) iris coloration in life, white background, flesh coloured towards the centre, fine brown reticulations; (18) melanophores present on dorsal surfaces of hands and feet and at the base of Finger IV, Toe IV, and Toe V; (19) males call from upper side of leaves; advertisement call unknown; (20) fighting behaviour unknown; (21) egg masses and parental care unknown; (22) tadpoles undescribed; (23) snout-vent length in adult males 29.1–31.2 mm (n = 2), females unknown.

Diagnosis. Centrolene camposi sp. nov. differs from all other glassfrogs, except C. altitudinale, C. buckleyi, C. heloderma, C. hesperia, C. lemniscata, and C. venezuelense by having a combination of the following characters: absence of dentigerous process of vomer, sloping snout in lateral view, light labial stripe, humeral spine in adult males, parietal peritoneum covered by iridophores, visceral peritonea translucent (except pericardium), ulnar and tarsal ornamentation, green bones. Centrolene altitudinale differs from C. camposi sp. nov. by having (characters of C. camposi sp. nov. in parentheses) truncate snout in dorsal view (rounded to subacuminate), tympanic annulus ½ visible (tympanic annulus barely visible), green dorsum with white dorsal spots in life (uniform green dorsum with light green warts); row of small, non-connected, enamelled tubercles on outer borders of hand, ulna, and tarsus (enamelled folds). Centrolene buckleyi and C. venezuelense differ by having ulnar and tarsal folds low or absent (elevated and thick ulnar and tarsal folds). Centrolene heloderma differs by having pustular dorsal skin (shagreen with dispersed warts), tympanic annulus completely visible (tympanic annulus barely visible), grey lavender dorsum in preservative (lavender), outer tarsal fold with low white tubercles (enamelled fold without tubercles), and humeral spine distinctly projected from arm (humeral spine curved towards arm). Centrolene hesperia differs by having weakly truncate snout in dorsal view (rounded to subacuminate), less hand webbing, II 2+–3½ III 3−–2½ IV (II (2−–2)–3+ III 2½–2+ IV). Centrolene lemniscata differs by having round snout in dorsal and lateral views (sloping in lateral view), arms and legs lacking dermal folds (present), and a white lateral stripe extending from arm insertion to groin. The second new species of Centrolene described in this work differs from C. camposi sp. nov. (characters of the later in parenthesis) by its round snout in lateral view (sloping), thin yellowish labial stripe (thick, white labial stripe), row of white tubercles between lip and arm insertion (white tubercles absent), yellowish line between arm insertion and groin (faint white line between lip and anterior ¼ of body), warts and spicules on dorsum with same colour as surrounding dorsal surfaces (warts and spicules on dorsum lighter than surrounding dorsal surfaces). Centrolene condor, sister species of C. camposi sp. nov., differs by having a green dorsum with abundant yellowish–white flecks and abundant dark flecks (bright uniformly green dorsum, sometimes with dark flecks); iris cream–yellow with fine dark reticulation (white background, flesh coloured towards the centre, fine brown reticulations); and vomerine teeth present (absent).

Description of the holotype. Adult male, moderate-sized, SVL = 29.1 mm (Figs. 2–5). Head distinct, wider than long, and wider than body; HW/HL = 1.10, HW/SVL = 0.38, HL/SVL = 0.35. Snout short, EN/HL = 0.24; nostrils slightly elevated, producing a shallow depression in the internarial area, loreal region concave; canthus rostralis rounded; lips flared. Small-size eyes, ED/HL = 0.31, directed anterolaterally at about 50° from midline, interorbital area wider than eye diameter, IOD/ED = 1.71, EN/ED = 0.77, EN/IOD = 0.58. Tympanic annulus oriented dorsolaterally, weak supratympanic fold above upper portion of tympanum and extending down to shoulder. Dentigerous processes of vomers absent; choanae rounded, large; tongue rounded, indented posteriorly; vocal slits present, extending from anterior base of tongue to angles of jaws.

Skin of dorsal surfaces of head, body and limbs shagreen with dispersed low warts, some warts are non-clustered translucent spicules, and the skin is covered by non-clustered microspicules, infratympanic area with spicules. Skin of ventral surfaces of body granular, on throat, chest, and limbs fairly granular. Cloacal opening directed posteriorly at upper level of thighs, distinct enamelled cloacal sheath; subcloacal area enamelled and granular, with a pair of large, round, flat subcloacal warts on ventral surfaces of thighs below vent.

Upper arm thin, forearm moderately robust. Humeral spine present and externally visible, but not piercing the skin. Relative lengths of fingers III > IV > II > I; finger discs wider than the adjacent phalanx, nearly truncate; disc on third finger about the same size than those on toes, ED/Fin3DW = 1.48; subarticular tubercles rounded and elevated, supernumerary tubercles present; palmar tubercle large, rounded, elevated; thenar tubercle elliptic. Concealed prepollex, unpigmented nuptial excrescences present, Type I on dorsolateral side of thumbs.

Hind limbs slender; TL/SVL = 0.54, FL/SVL = 0.49. Inner metatarsal tubercle large and elliptical; outer metatarsal tubercle indistinct. Subarticular tubercles rounded and low, supernumerary tubercles small, rather indistinct. Toe discs bluntly truncate, no papillae on tip of disc of toes.

Colouration of holotype in life. (Figs. 2 and 3) Bright green dorsal colouration, with some warts slightly lighter green; thick, yellowish-white labial stripe continuing into a faint yellowish line between lip and anterior ¼ of body; yellowish-green flanks and hidden surfaces of limbs; enamelled metacarpal, ulnar, metatarsal, and tarsal folds; yellowish white venter. Iris with grey background, fleshed coloured towards the centre, fine brown reticulations. Discs orange to red in Fingers II, III and IV on the left hand, Fingers II and IV on the right hand, and Toe V on both feet. Yellowish green webbing between fingers and toes. Bones green.

Colouration of holotype in ethanol. (Figs. 4 and 5) Lavender dorsum with translucent spicules; enamelled labial stripe continuing into a faint enamelled line between lip and anterior ¼ of body; flanks lighter lavender than dorsal surfaces; faint enamelled metacarpal, ulnar, metatarsal, and tarsal folds; venter cream. Melanophores present on dorsal surfaces of hands and feet and at the base of Finger IV, Toe IV, and Toe V. Parietal peritoneum white, iridophores covering 2/3 the parietal peritoneum; pericardium covered by iridophores, all other visceral peritonea clear.

Measurements in mm: Measurements of the holotype are followed by those of the paratype in parentheses: SVL = 29.1 (31.2), HL = 10.1 (11.1), HW = 11.1 (11.5), IOD = 5.3 (5.7), ED = 3.1 (3.4), IND = 2.7 (2.9), EN = 2.4 (2.4), TD = 1.0 (1.2), TL = 15.8 (17.0), FL = 14.3 (14.8), HAL = 11.1 (11.6), Fin3DW = 2.1 (2.2).

Variation. Morphologically the paratype is very similar to the holotype, except for its snout subacuminate in dorsal view. Morphometric variation is reported in the previous section. The paratype shows dorsal warts lighter green than the holotype, almost looking like bright yellowish green dots, and has some dorsal dark flecks. The enamelled line on the anterior ¼ of the body is thinner than in the holotype.

Etymology. The specific name of this new taxon is an eponym in honour of Felipe Campos-Yánez, a distinguished Ecuadorian zoologist, free thinker, and passionate conservationist. His biological collections are deposited in the country’s main museums, and he has left a legacy of contributions to biodiversity conservation throughout his 30 years of professional career.

Distribution and Natural History. Centrolene camposi sp. nov. is known only from its type locality in the province of Azuay, near the border with the province of El Oro, on the southwestern slopes of the Cordillera Occidental of the Andes of Ecuador (Fig. 7), inhabiting montane evergreen forests at 2,900 m elevation. This ecosystem is characterized by trees greater than 15 m in height and densely loaded with epiphytes, such as bromeliads, mosses, and orchids. Both individuals of C. camposi sp. nov. were found together with C. ericsmithi sp. nov., in a steep creek. Centrolene camposi sp. nov. is also sympatric with Pristimantis allpapuyu Yánez-Muñoz, Sánchez-Nivicela & Reyes-Puig, 2016, four undescribed species of Pristimantis, and one Gastrotheca.

Figure 7 Map of Ecuador showing the type locality of the new species of Centrolene.

Type locality of Centrolene camposi sp. nov. and C. ericsmithi sp. nov. (black star), and the known localities of C. condor (grey squares), sister species to C. camposi.

Centrolene ericsmithi sp. nov.

LSID urn:lsid:zoobank.org:act:BA14CDBB-9BEB-4245-889F-ADB975775E74

(Figs. 2–6)

Centrolene heloderma Yánez-Muñoz (2015)

Centrolene sp. 1. Bejarano-Muñoz, Sánchez-Nivicela & Yánez-Muñoz (2019)

Proposed Spanish common name: Rana de Cristal de Smith

Proposed English common name: Smith’s Glassfrog

Holotype. DHMECN 11406 (field number 3546), adult male (Figs. 2 and 3) from La Enramada (3.1628°S; 79.5886°W, 2,950 m), provincia de Azuay, República del Ecuador, collected by J. Sánchez-Nivicela on 31 March 2015.

Diagnosis. Centrolene ericsmithi sp. nov. is distinguished from all other Centrolenidae by the following combination of characters: (1) dentigerous process of vomer absent; (2) snout round in dorsal and lateral views; (3) tympanic annulus barely visible, lower ¾ visible, tympanic membrane coloured as dorsal skin, supratympanic fold present and low; (4) dorsal skin shagreen with dispersed spicules, and covered by microspicules; (5) ventral skin granular, subcloacal area enamelled, strongly granular with two slightly larger subcloacal warts and enamelled cloacal sheath; (6) parietal peritoneum white, iridophores covering ½ parietal peritoneum (condition P3); pericardium covered by iridophores, all other visceral peritonea clear (condition V1); (7) liver lobed and hepatic peritoneum clear (lacking iridophore layer, condition H0); (8) adult males with projecting humeral spine; (9) basal webbing between fingers I and III, moderate webbing between fingers III and IV, III 2⅔ –2½ IV; (10) toe webbing I 2––2+ II 1––2⅓ III 2––2½ IV 2½–2– V; (11) enamelled metacarpal area without fold continuing with low, slightly elevated, enamelled ulnar fold; low, enamelled metatarsal and tarsal fold; low tarsal fringe on inner tarsus; (12) nuptial excrescences type I; concealed prepollex; (13) Finger I shorter than Finger II; (14) diameter of eye larger than width of disc on Finger III; (15) colour in life, bright green dorsum, thin yellowish labial stripe continuing with a row of white tubercles towards arm insertion, yellowish line between arm insertion and groin, enamelled metacarpal area, enamelled ulnar, metatarsal and tarsal fold, bones green; (16) colour in preservative, lavender dorsum with translucent spicules, enamelled labial stripe, enamelled line between arm insertion and groin; (17) iris coloration in life, flesh colour background, fine brown reticulations; (18) few melanophores present on dorsal surfaces of hands and feet and at the base of Finger IV, Toe IV, and Toe V; (19) males call from upper side of leaves; advertisement call unknown; (20) fighting behaviour unknown; (21) egg masses and parental care unknown; (22) tadpoles undescribed; (23) snout-vent length (SVL) in adult male 27.3 mm (n = 1), females unknown.

Comparisons. Centrolene ericsmithi sp. nov. differs from all other glassfrogs, except from C. altitudinale, C. buckleyi, C. heloderma, C. hesperia, C. lemniscata, C. sabini, and C. venezuelense by having a combination of the following characters: absence of dentigerous process of vomer, light labial stripe, uniform green dorsum, humeral spine in adult males, parietal peritoneum covered by iridophores, visceral peritonea translucent (except pericardium), ulnar and tarsal ornamentation, green bones. Centrolene altitudinale differs from C. ericsmithi sp. nov. by having (characters of C. ericsmithi sp. nov. in parentheses) truncate snout in dorsal view (rounded), tympanic annulus ½ visible (tympanic annulus barely visible), dorsum shagreen with small spicules (shagreen with large spicules), green dorsum with white dorsal spots in life (uniform green dorsum); row of small, non-connected, enamelled tubercles on outer borders of ulna and tarsus (enamelled folds). Centrolene buckleyi and C. venezuelense differ by having sloping snout (round), supratympanic fold moderately heavy (low), outer tarsal fold absent (present); iris with a horizontal brown stripe (brown stripe absent). Centrolene camposi sp. nov. differs by having sloping snout in lateral view (round), thick, white labial stripe (thin yellowish labial stripe), absence of row of white tubercles between lip and arm insertion (present), faint white line between lip and anterior ¼ of body (yellowish line between arm insertion and groin), warts and spicules on dorsum lighter than surrounding dorsal surfaces (warts and spicules on dorsum with same colour as surrounding dorsal surfaces). Centrolene heloderma differs by having pustular dorsal skin (shagreen with dispersed spicules), tympanic annulus completely visible (tympanic annulus barely evident), grey lavender dorsum in preservative (lavender); outer tarsal fold with low white tubercles (enamelled fold without tubercles), and humeral spine distinctly projected from arm (humeral spine curved towards arm). Centrolene hesperia differs by having weakly truncate snout in dorsal view (rounded) and white labial stripe continuous with stripe along the flanks to the groin (labial stripe separate from body line by a row of tubercles). Centrolene lemniscata differs by arms and legs lacking dermal folds (present) and white labial stripe continues along the body to the groin (labial stripe separate from body line by a row of tubercles). Centrolene sabini differs by having sloping snout in lateral view (round), dorsum green with yellowish-green spots and patches (uniformly green), white labial stripe continuous with stripe along the flanks (labial stripe separate from body line by a row of tubercles), and strongly protruding nostrils (not strongly protruding). Centrolene lynchi differs by having snout truncate to sloping in lateral view (round), dorsal skin shagreen in males and females, males with low, white warts, and spicules and spiculated warts on sides of head (dorsal skin shagreen with dispersed spicules); dorsum dull green with minute yellowish–white warts and small diffuse black spots (green dorsum), tarsal fold absent (present), nuptial pad Type II (Type I), and humeral spine distinctly projected from arm (humeral spine curved towards arm). Molecular analyses clearly differentiate C. ericsmithi sp. nov. from morphologically similar species found in the Andes.

Description of the holotype. Adult male, moderate-sized, SVL = 27.3 mm (Figs. 2–5). Head slightly distinct, wider than long, and wider than body; HW/HL = 1.06, HW/SVL = 0.33, HL/SVL = 0.31. Snout short, EN/HL = 0.21; nostrils slightly elevated, producing a shallow depression in the internarial area, loreal region concave; canthus rostralis rounded; lips not flared. Small-size eyes, ED/HL = 0.33, directed anterolaterally at about 50° from midline, interorbital area wider than eye diameter, IOD/ED = 1.43, EN/ED = 0.64, EN/IOD = 0.70. Tympanic annulus oriented dorsolaterally, weak supratympanic fold above upper portion of tympanum and extending down to shoulder. Dentigerous processes of vomers absent; choanae squarish, large; tongue rounded, indented posteriorly; vocal slits present, extending from anterior base of tongue to angles of jaws.

Skin of dorsal surfaces of head, body and limbs shagreen, covered by non-clustered translucent spicules, spicules more concentrated on body surfaces, infratympanic area with few, slightly enlarged spicules. Dorsal surfaces with non-clustered microspicules. Skin of ventral surfaces of body granular, on throat, chest, and limbs fairly shagreen. Cloacal opening directed posteriorly at upper level of thighs, distinct enamelled cloacal sheath; subcloacal area enamelled and granular, with a pair of slightly large, round, flat subcloacal warts on ventral surfaces of thighs below vent.

Upper arm thin, forearm slightly robust. Humeral spine present and externally visible, but not piercing skin. Relative lengths of fingers III > IV > II > I; finger discs wider than the adjacent phalanx, truncate; disc on third finger about the same size than those on toes, ED/Fing3DW = 1.87; subarticular tubercles rounded and elevated, supernumerary tubercles present, small, and flat; palmar tubercle large, rounded, elevated; thenar tubercle elliptic. Concealed prepollex, unpigmented nuptial excrescences present, Type I on dorsolateral side of thumbs.

Hind limbs slender; TL/SVL = 0.55, FL/SVL = 0.47. Inner metatarsal tubercle large and elliptical; outer metatarsal tubercle indistinct. Subarticular tubercles rounded and low, supernumerary tubercles small and flat, rather indistinct. Toe discs bluntly truncate, no papillae on tip of disc of toes.

Colouration of holotype in life. (Figs. 2 and 3) Bright, uniform green dorsum, thin yellowish labial stripe continuing with a row of white tubercles towards arm insertion, yellowish line between arm insertion and groin, enamelled metacarpal area, enamelled ulnar, metatarsal, and tarsal fold, yellowish white venter. Iris flesh colour background, fine brown reticulations. Fingers, toes, and membrane yellowish green. Bones green.

Colouration of holotype in ethanol. (Figs. 4 and 5) Lavender dorsum with translucent spicules, enamelled labial stripe, enamelled line between arm insertion to groin. Faint enamelled metacarpal area, faint enamelled ulnar, metatarsal, and tarsal fold. Few melanophores present on dorsal surfaces of hands and feet and at the base of Finger IV, Toe IV, and Toe V. Parietal peritoneum white, iridophores covering ½ parietal peritoneum; pericardium covered by iridophores, all other visceral peritonea clear.

Measurements of the holotype: SVL = 27.3, HL = 8.5, HW = 9.0, IOD = 4.0, ED = 2.8, IND = 2.5, EN = 1.8, TD = 0.9, TL = 15.1, FL = 12.8, HAL = 9.4, Fin3DW = 1.5.

Etymology. The specific name of this new taxon is an eponym in honour of Eric Nelson Smith, U.S. herpetologist and curator of the amphibian and reptile collections at The University of Texas at Arlington. Eric is a prodigious collector that has described more than 60 new species of amphibians and reptiles from the most remote corners of the planet for almost three decades. This is a small recognition of his extensive contributions.

Distribution and Natural History. Centrolene ericsmithi sp. nov. is currently known only from its type locality in the province of Azuay, on the southwestern slopes of the the Cordillera Occidental of the Andes of Ecuador (Fig. 7). Centrolene ericsmithi sp. nov. and C. camposi sp. nov. are syntopic. Both new species vocalized during the samplings in March 2015, but unfortunately, no recordings were taken. Other relevant information is described in the section corresponding to C. camposi.

Discussion

Due to their sympatry, with overlapping microhabitat occupancy, Yánez-Muñoz (2015) initially assumed that the three specimens of Centrolene from La Enramada, province of Azuay, belonged to the same species and were phylogenetically close to C. heloderma. However, the phylogenetic reconstruction showed they were two different, syntopic, not closely related lineages, C. camposi sp. nov. and C. ericsmithi sp. nov.

The two new species of Centrolene described herein inhabit the montane evergreen forests in the south-western Andes of Ecuador. Centrolene camposi shows an unusual biogeographic pattern because it is sister to a species from the opposite versant of the Andes (Fig. 7). Relatively low genetic distances separate both species (average 1.08% for gene 12S), suggesting a recent divergence. The Andes are a formidable dispersal barrier for amphibians, resulting in almost entirely different amphibian communities on opposite versants of the Andes of Ecuador, despite having ecologically similar forests. In Centrolenidae, only two other sister species occur on opposite versants of the Andes, T. amelie (Cisneros-Heredia & Meza-Ramos, 2007) + Teratohyla pulverata (Peters, 1873) and Cochranella granulosa (Taylor, 1949) + C. resplendens (Lynch & Duellman, 1973; Guayasamin et al., 2020). Teratohyla amelie and T. pulverata diverged 15 My ago, suggesting vicariant speciation due to the Andean uplift. Cochranella granulosa + C. resplendens are old lineages, diverging over 7 My ago (Guayasamin et al., 2020). In contrast, C. camposi + C. condor likely diverged much more recently because their genetic distances are at the lower end for species pairs within Centrolenidae. Species of the genus Centrolene occupy much higher elevations than other species of Centrolenidae; thus, trans-Andean distributions might have been possible until more recent geological periods. This unusual biogeographic pattern suggests a unique combination of topography and environmental history in the Andes of southern Ecuador. The pattern of southwestern Andean clades in Ecuador and Peru having a closer relationship with eastern Andean clades should be much more common in species with high dispersal ability like spiders (e.g., Gasteracantha cancriformis (Linnaeus, 1758); Salgado-Roa et al., 2022) and birds (e.g., Pachyramphus spp., Musher & Cracraft, 2018).

Centrolene condor is endemic to the Cordillera del Condor, a sub-Andean mountain range running parallel to the southeastern Andes of Ecuador, about 140 km W from the type locality of C. camposi (Fig. 7) (Cisneros-Heredia & Morales-Mite, 2008; Almendáriz & Batallas, 2012; Guayasamin et al., 2020). Centrolene sabini is only known from the Kosñipata valley in the southeastern Andes of Peru, more than 1,400 km south of the type locality of C. ericsmithi. (Catenazzi et al., 2012; Catenazzi, 2017). The undescribed Centrolene sp. [Ca1] was collected in the province of Zamora-Chinchipe, in the southeastern Andes of Ecuador, about 120 km W from the type locality of C. ericsmithi (Amador et al., 2018). The undescribed Centrolene sp. [Ca1] is more closely related to the geographically distant C. sabini than to the geographically close C. ericsmithi sp. nov., a relationship that counters the usual isolation by distance pattern of genetic differentiation among populations of a single species. Lack of consistency with isolation by distance suggests the existence of reproductive barriers between the three populations (i.e., the populations belonging to three species).

The diversification and adaptation of a high diversity of vertebrates in southwestern Ecuador, specifically in the province of Azuay and El Oro, is apparently due to the topographic complexity of the area, with the presence of the River Jubones basin and the Gulf of Guayaquil, the Andes and its proximity to the Pacific coast, and the biogeographic influence of different climatic zones (INABIO, 2015; Arteaga et al., 2016, 2017, 2018; Yánez-Muñoz, Sánchez-Nivicela & Reyes-Puig, 2016; Torres-Carvajal et al., 2020; Brito et al., 2022). Evidence accumulated in several clades of small vertebrates suggests that the River Jubones basin might be an important isolation barrier between lineages north and south of the Andes of Ecuador, including frogs of the genera Pristimantis, Elachistochleis, Hyloscirtus and Nymphargus, lizards Anadia, Enyalioides and Stenocercus, and snakes Atractus, Dipsas and Leptodeira (Torres-Carvajal, 2007; Passos, Cisneros-Heredia & Salazar-V, 2007; Cisneros-Heredia & Yánez-Muñoz, 2007b; Passos et al., 2012; INABIO, 2015; Arteaga et al., 2016, 2018; Yánez-Muñoz, Sánchez-Nivicela & Reyes-Puig, 2016; Sánchez-Nivicela et al., 2018, 2019, 2020; Betancourt et al., 2018; Guayasamin et al., 2020; Torres-Carvajal et al., 2020; Yánez-Muñoz et al., 2021).

The Andes of southern Ecuador show high geologic, geographic, and environmental heterogeneity (Gentry, 1982; Duque-Caro, 1990; Veblen, Young & Orme, 2015; Morrone, 2017). The combination of these factors has fostered the evolution of a complex and rich biological diversity, with several local hotspots concentrating high levels of endemism (Chapman, 1917, 1926; Gentry, 1982; Cracraft, 1985; Duellman, 1988; Dodson & Gentry, 1991; Morrone, 2014, 2015, 2017). Unfortunately, its biodiversity remains poorly studied and heavily threatened by unceasing habitat loss, degradation, and fragmentation due to legal and illegal logging, expansion of the agricultural frontier, and mining activities (MAE, 2012, 2015; MAE et al., 2013; Sierra, 2013). The remnants of native montane forests in the provinces of Azuay and El Oro are restricted and scarce. Even though we have carried out extensive surveying of amphibians in the region, no more individuals or localities of the new species have been reported, so we consider that both species should be assigned to the IUCN Red List category Data Deficient at the national and global levels (Ortega-Andrade et al., 2021). The discovery of these new species of anurans in small Andean remnants evidences the need to carry out urgent conservation actions, to avoid the collapse of these ecosystems in southwestern Ecuador (INABIO, 2015; Ortega-Andrade et al., 2021).

Conclusions

We provide congruent lines of evidence supporting the diagnosis and description of two new species of Centrolene from the southwestern high Andes of Ecuador. These new species were found sympatric in a steep creek covered by montane evergreen forest at 2,900 m at La Enramada, province of Azuay, near the border with the province of El Oro, on the southwestern slopes of the Andes of Ecuador. Our phylogeny places C. camposi sp. nov. as the sister species of C. condor and C. ericsmithi as the sister to a clade composed of C. sabini and an undescribed species of Centrolene from southeastern Ecuador.

The Andes have historically been a formidable dispersal barrier for amphibians, resulting in distinct amphibian communities on opposite versants of the Andes, despite having similar ecosystems. The unusual biogeographic pattern observed in the clade composed of C. camposi and C. condor suggests a unique combination of topography and environmental history in the Andes of southern Ecuador. In addition, the River Jubones basin is an important isolation barrier for small vertebrates in the western Andes of Ecuador. The study highlights the importance of studying geologic and biogeographic events’ role in shaping species’ diversity and distribution.

Supplemental Information

Supplemental Information 1 Matrix used for phylogenetic analyses.

Click here for additional data file.

We express our gratitude to the Gobierno Autónomo Provincial de El Oro for their support during field and lab work through the project “Amphibians, Reptiles and Birds of the El Oro Province” executed by INABIO. We thank the following people for their support at their respective institutions: Diego Inclán, Francisco Prieto, César Garzón, and Miguel Urgiles-Merchán (INABIO); William E. Duellman, Linda Trueb, Juan M. Guayasamin and Elisa Bonaccorso (KU); George Zug, Roy McDiarmid, Ron Heyer, Robert Reynolds, Kenneth A. Tighe, Steve W. Gotte, Carole C. Baldwin, and Mary Sangrey (USNM); Carolina Reyes-Puig and Emilia Peñaherrera (Museo de Zoología ZSFQ, Universidad San Francisco de Quito). Emilia Peñaherrera provided help with the map. Mario Yánez-Muñoz is especially thankful to Patricia Bejarano-Muñoz and Juan Carlos Sánchez-Nivicela to Lorena Orellana for their continuous support and patience during the long days in the field. Diego F. Cisneros-Heredia thanks María Elena Heredia, Laura Heredia, and Jonathan Guillemot for their constant support. We are thankful to Evan Twomey and Diego Armijos-Ojeda for their encouraging words and detailed comments.

Additional Information and Declarations

Competing Interests

Author Contributions

Animal Ethics

Field Study Permissions

DNA Deposition

Data Availability

New Species Registration

Santiago R. Ron is a PeerJ Academic Editor. The authors declare that they have no competing interests.

Diego F. Cisneros-Heredia conceived and designed the experiments, performed the experiments, analyzed the data, prepared figures and/or tables, authored or reviewed drafts of the article, and approved the final draft.

Mario H. Yánez-Muñoz conceived and designed the experiments, performed the experiments, analyzed the data, prepared figures and/or tables, authored or reviewed drafts of the article, and approved the final draft.

Juan C. Sánchez-Nivicela conceived and designed the experiments, performed the experiments, authored or reviewed drafts of the article, and approved the final draft.

Santiago R. Ron conceived and designed the experiments, performed the experiments, analyzed the data, prepared figures and/or tables, authored or reviewed drafts of the article, and approved the final draft.

The following information was supplied relating to ethical approvals (i.e., approving body and any reference numbers):

Our study was authorised under framework contracts for access to genetic resources MAE-DNB-CM-2016-0045 and MAE-DNB-CM-2019-0120, issued by the Ministerio del Ambiente del Ecuador.

The following information was supplied relating to field study approvals (i.e., approving body and any reference numbers):

Our study was authorised under framework contracts for access to genetic resources MAE-DNB-CM-2016-0045 and MAE-DNB-CM-2019-0120, issued by the Ministerio del Ambiente del Ecuador.

The following information was supplied regarding the deposition of DNA sequences:

The newly generated DNA sequences used in genetic analyses in this study are available at GenBank: OQ225626, OQ248672, OQ248679, OQ248668, OQ225627, OQ248673, OQ248680, OQ248669, OQ225628, OQ248674, OQ248682, OQ248670, OQ225629, OQ225616, OQ248675, OQ248681, OQ248676, OQ248683, OQ225630, OQ248677, OQ248671, OQ225617, OQ248678.

The following information was supplied regarding data availability:

All raw data is provided directly in the manuscript.

The following information was supplied regarding the registration of a newly described species:

Publication LSID: urn:lsid:zoobank.org:pub:A2A88B00-DA2C-443E-BC8B-9922980F8789

Centrolene camposi LSID urn:lsid:zoobank.org:act:868316B5-0ED5-4A21-AE3A-0488D98E418B

Centrolene ericsmithi LSID urn:lsid:zoobank.org:act:BA14CDBB-9BEB-4245-889F-ADB975775E74

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
