# Peer review of "Two new syntopic species of glassfrogs (Amphibia, Centrolenidae, Centrolene) from the southwestern Andes of Ecuador"

_PeerJ, doi:10.7717/peerj.15195_

## Round 0.1 · original submission · Minor Revisions

Please provide a revised version, considering the suggestions made by the reviewers, and send it back as soon as possible.

·

Basic reporting

The article meets the criteria of scientific validity and suitability. Check the document for British English. I understand that PeerJ requests American English (Colouration vs. Color)

Experimental design

I have no comments

Validity of the findings

I have no comments

Additional comments

Reviewing the phylogenetic tree, the specimen QCAZ44896 in Guayasamin et al. 2020 is cited like Centrolene sp Ca04, but in your article this specimen is Centrolene condor. ¿How was the decision made?

I know the Centrolene condor sequences does not exist

·

Basic reporting

This manuscript was a pleasure to read and made for a very easy review assignment. First, the two species are clearly new, the first of which is easily distinguished from its sister species (Ce. condor) by the presence/absence of vomerine teeth, among other characters. The second species is sister to a species from southern Peru, which differs in a number of characters. The manuscript is impeccably written, with crystal clear diagnosis sections. The manuscript also includes substantial new sequence data, which not only helps resolve the relationships of the new species, but provides a substantial contribution to glassfrog systematics. The only weaknesses were the absence of bioacoustic data, and low sample sizes for both species. Still, the descriptions of both species are justified and well-supported.

Most of my comments are minor and should be easy to address in a revision.

Lines 288-289, it is stated that C. condor has dorsal color green with abundant yellowish-white flecks and dark flecks, versus uniform green in C. camposi sp. nov. However, in the variation section, the paratype of the new species is described as having light green dorsal warts/yellowish dots, as well as dark dorsal flecks, which conflicts with what is written in Comparisons. The Comparisons section should be updated to account for this variation.

Line 49: Morphological synapomorphies. Because if it is a well-supported clade there surely must be at least a handful of molecular ones.

The punctuation around lines 99-112 is hard to follow. For example: Ecuador: Bolivar: DHMECN 0866-67, Guanujo; Carchi:

It seems that locality would be Ecuador: Bolivar: Guanujo, so putting the third-level of locality specification *after* the specimen numbers is confusing. Better would be something like

Ecuador: Bolivar: Guanujo (DHMECN 0866-67)

I'm honestly not sure I'm even getting this right. For Pinchicha province, USNM 288423 uses a comma to separate what I guess is the more precise locality (Quito). The next entry uses a colon (: 8.5 km by road).

Basically it is just way more confusing than necesssary. Maybe just set identical levels in the same font, e.g. species = bold italic, country = all caps, Department = Underlined, specimen numbers in parentheses after the most precise locality description.

Line 137: should be "live specimens"

Line 142: "affinities" is probably not the best word choice. Relationships?

Line 212: Is this the average p-distance across all genes? Above, it was specified which genes, but here, it seems that average values are given.

Line 234: Following ICZN, this would be the definition, not the diagnosis:
Definition, n.: A statement in words that purports to give those characters which, in combination, uniquely distinguish a taxon [Arts. 12, 13].

Diagnosis, n.: A statement in words that purports to give those characters which differentiate the taxon from other taxa with which it is likely to be confused.

Line 255: change to "flesh coloured". Same comment on line 290.

Line 357: sentence fragment, consider revising, e.g. joining with previous sentence "(Fig. 7), inhabiting montane..."

Line 477: Period after "green".

Line 491: Sentence fragment, revise. Also, change to "...that has described 50 new species..." Further, do you mean exactly 50 species? More than 50 species? Species of what? Amphibians and reptiles? "Extensive career" somewhat awkward, consider "extensive contributions"

Line 498: Maybe mention something like "but unfortunately, no recordings were taken", to clarify that bioacoustic data still does not exist for either species.

Lines 517-526: Another point worth mentioning is that Centrolene species occupy much higher elevations than the other species mentioned here, and thus trans-Andean distributions may have been possible up until much more recently.

The Conclusions section, except the first sentence, is not conclusions at all (ie., does not re-state or summarize anything earlier in the manuscript) but is in fact additional discussion. I would suggest simply removing the Conclusions heading and add these paragraphs to the end of the discussion.

Line 551: change to "Jubones River"

Line 535: The last two letters in Centrolene are not italicized

Line 592: Centrolene condor not in italics

Experimental design

see box 1 for all comments

Validity of the findings

see box 1 for all comments

Additional comments

see box 1 for all comments

---

## Round 0.2 · Minor Revisions

The authors have provided a fully revised version of their manuscript, that I have read it carefully and find apropriate. Herein I enclose a PDF with a few minor corrections. The authors are requested to send a final version after checking these corrections.

---

## Round 0.3 · accepted · Accept

The authors have addressed the comments made by the reviewers, so I accept this revised version.